An explicit-solvent conformation search method using open software

Gaalswyk Kari
http://orcid.org/0000-0002-0205-952X Rowley Christopher N. crowley@mun.ca
Department of Chemistry, Memorial University of Newfoundland , St. John’s, Newfoundland and Labrador , Canada
Silva Pedro
Electronic publication date: 2016 May 31
Publication date: 2016
Volume: 4
Electronic Location ID: e2088
Received 2016 Apr 5; Accepted 2016 May 6
Copyright: © 2016 Gaalswyk & Rowley
Copyright year: 2016
Copyright holder: Gaalswyk & Rowley
License: This is an open access article distributed under the terms of the Creative Commons Attribution License, which permits unrestricted use, distribution, reproduction and adaptation in any medium and for any purpose provided that it is properly attributed. For attribution, the original author(s), title, publication source (PeerJ) and either DOI or URL of the article must be cited.
License URL: https://creativecommons.org/licenses/by/4.0/

Keywords: Conformation search, Explicit solvent, Cluster analysis, Replica exchange molecular dynamics

Funding: NSERC of Canada Memorial University The authors received funding from the NSERC of Canada through the Discovery Grant program (Application 418505-2012). Kari Gaalswyk received funding from Memorial University. Computational resources were provided by Compute Canada (RAPI: djk-615-ab) through the Calcul Quebec and ACEnet consortia. The funders had no role in study design, data collection and analysis, decision to publish, or preparation of the manuscript.

==============================
Computer modeling is a popular tool to identify the most-probable conformers of a molecule. Although the solvent can have a large effect on the stability of a conformation, many popular conformational search methods are only capable of describing molecules in the gas phase or with an implicit solvent model. We have developed a work-flow for performing a conformation search on explicitly-solvated molecules using open source software. This method uses replica exchange molecular dynamics (REMD) to sample the conformational states of the molecule efficiently. Cluster analysis is used to identify the most probable conformations from the simulated trajectory. This work-flow was tested on drug molecules α-amanitin and cabergoline to illustrate its capabilities and effectiveness. The preferred conformations of these molecules in gas phase, implicit solvent, and explicit solvent are significantly different.

Introduction

Many molecules can exist in multiple conformational isomers. Conformational isomers have the same chemical bonds, but differ in their 3D geometry because they hold different torsional angles (Crippen & Havel, 1988). The conformation of a molecule can affect chemical reactivity, molecular binding, and biological activity (Struthers, Rivier & Hagler, 1985; Copeland, 2011). Conformations differ in stability because they experience different steric, electrostatic, and solute-solvent interactions. The probability, p, of a molecule existing in a conformation with index i, is related to its relative Gibbs energies through the Boltzmann distribution, pi=exp(−ΔGi/kBT)Σjexp(−ΔGj/kBT)(1)

where kB is the Boltzmann constant, T is the temperature, and ΔG is the relative Gibbs energy of the conformation. The denominator enumerates over all conformations.

Alternatively, the probability of a conformation can be expressed in classical statistical thermodynamics in terms of integrals over phase space, pi=∫iexp(−V(r)/kBT)dr∫exp(−V(r)/kBT)dr(2)

The integral over configurational space in the numerator is restricted to coordinates corresponding to conformation i. The denominator is an integral over all configurational space. V(r) is the potential of the system at when the atoms hold coordinates r.

Computational chemistry has enabled conformational analysis to be performed systematically and quantitatively with algorithms to generate different conformations and calculate their relative stability. Automated conformational search algorithms can generate possible conformations, and molecular mechanical or quantum methods can determine their relative energies.

Conformational search methods can be classified as either exhaustive/systematic or heuristic. Exhaustive methods scan all, or a significant portion of the configuration space. Subspaces corresponding to high energy structures can be eliminated without a loss in quality using a priori knowledge regarding the structure of the configuration space to be searched (Christen & van Gunsteren, 2008). These methods are usually limited to small molecules due to the computational cost of searching so much of the configuration space. Heuristic methods generate a representative set of conformations by only visiting a small fraction of configuration space (van Gunsteren et al., 2006). These methods can be divided into non-step and step methods. Non-step methods generate a series of system configurations that are independent of each other. Step methods generate a complete system configuration in a stepwise manner by a) using configurations of molecular fragments, or b) using the previous configuration (Christen & van Gunsteren, 2008).

Solvent effects

A solvent can also affect the conformation of a molecule by effects like solvent-solute hydrogen bonding, dipole-dipole interactions, etc. (Christen & van Gunsteren, 2008). Incorporating the effect of solvation can complicate conformation searches. It is common to perform a conformation search in the gas phase, neglecting solvent effects altogether. Alternatively, the solvent can be included in the simulation either implicitly or explicitly.

Implicit models approximate the solvent as a dielectric continuum interacting with the molecular surface (Anandakrishnan et al., 2015). Depending on the model used, the computational cost of calculating the solvation can be modest, allowing solvation effects to be included in the conformation search. A common and efficient implicit solvent method used with molecular mechanical models is the Generalized Born Implicit Solvent (GBIS) method (Bhandarkar et al., 2015). A limitation of this type of model is that features like solute-solvent hydrogen bonding and solute-induced changes in the solvent structure are difficult to describe accurately when the solvent is described as a continuum.

Explicit solvation methods surround the solute with a number of solvent molecules that are represented as discrete particles. Provided that this model accurately describes solvent molecules and their interactions with the solute, some of the limitations in accuracy associated with implicit solvent models can be overcome. Although the accuracy of these models is potentially an improvement over continuum models, the inclusion of explicit solvent molecules presents challenges in conformation searches. Some conformational search algorithms that arbitrarily change dihedral angles cannot be used in an explicit solvent because an abrupt change in a solute dihedral angle can cause an overlap with solvent molecules.

A significant drawback of explicit solvent representations is that the computational cost of these simulations is increased considerably due to the additional computations needed to describe the interactions involving solvent molecules. Longer simulations are also needed to thoroughly sample the configurations of the solvent; the stability of each conformation is the result of a time average over an ensemble of possible solvent configurations (i.e., its Gibbs/Helmholtz energy), rather than the potential energy of one minimum-energy structure.

Previous work

Many conformational search methods have been developed. Sakae et al. (2015) used a combination of genetic algorithms and replica exchange. They employed a two point crossover, where consecutive amino acid residues were selected at random from each pair, and then the dihedral angles were exchanged between them. Superior conformations were selected using the Metropolis criterion, and these were then subjected to replica-exchange. Supady, Blum & Baldauf (2015) also used a genetic algorithm where the parents were chosen using a combination of three energy-based probability metrics.

One example of a systematic method is the tree searching method of Izgorodina, Lin & Coote (2007). The method optimizes all individual rotations, and then ranks their energies. It then eliminates those with relative energies greater than the second lowest energy conformer from the previous round, and performs optimizations on only the remaining subset. After a set number of rotations, the lowest ranked conformer is selected. Brunette & Brock (2008) developed what they called a model-based search, and compared it to traditional Monte Carlo. The model-based search characterizes regions of space as funnels by creation an energy-based tree where the root of the tree corresponds to the bottom of the funnel. The funnel structure illustrates the properties of the energy landscape and the sample relationships. Cappel et al. (2014) tested the effects of conformational search protocols on 3D quantitative structure activity relationship (QSAR) and ligand based virtual screening.

Perez-Riverol et al. (2012) developed a parallel hybrid method that follows a systematic search approach combined with Monte Carlo-based simulations. The method was intended to generate libraries of rigid conformers for use with virtual screening experiments.

Some methods have been extended to incorporate physical data. MacCallum, Perez & Dill (2015) developed a physics-based Bayesian computational method to find preferred structures of proteins. Their Modeling Employing Limited Data (MELD) method identifies low energy conformations from replica-exchange molecular dynamics simulations that are subject to biases that are based on experimental observations.

Conformation searches using molecular dynamics

Molecular dynamics (MD) simulations are a popular method for sampling the conformational space of a molecule. Equations of motion are propagated in a series of short time steps that generates a trajectory describing the motion of the system. These simulations are usually coupled to a thermostat to sample a canonical or isothermal–isobaric ensemble for the appropriate thermodynamic state. This approach is inherently compatible with explicit solvent models because the dynamics will naturally sample the solvent configurations. For a sufficiently long MD simulation, the conformational states of the molecule will be sampled with a probability that reflects their relative Gibbs/Helmholtz energies. This is in contrast to many conformational search methods that search for low potential energy conformations.

One of the limitations of MD is that very long simulations may be needed to sample the conformational states of a molecule with the correct weighting. This occurs because MD simulations will only rarely cross high barriers between minima, so a simulation at standard or physiological temperatures may be trapped in its initial conformation and will not sample the full set of available conformations.

Replica Exchange Molecular Dynamics (REMD) enhances the sampling efficiency of conventional MD by simulating multiple copies of the system at a range of temperatures. Each replica samples an ensemble of configurations occupied at its corresponding temperature. Periodically, attempts are made to exchange the configurations of neighboring systems (see Fig. 2). The acceptance or rejection of these exchanges is determined by an algorithm analogous to the Metropolis Monte Carlo algorithm, which ensures that each replica samples its correct thermodynamic distribution. This type of simulation is well suited for parallel computing because replicas can be divided between many computing nodes. Exchanges between the replicas are only attempted after hundreds or thousands of MD steps, so communication overhead between replicas is low compared to a single parallel MD simulation.

REMD simulations can sample the conformational space of a molecule more completely because the higher temperature replicas can cross barriers more readily. Analysis of the statistical convergence of REMD simulations has shown that when there are significant barriers to conformational isomerization, an REMD simulation of m replicas is more efficient than a single-temperature simulation running m times longer (Sugita & Okamoto, 1999). The lowest temperature replica is typically the temperature of interest. Exchanges allow each replica to be simulated at each temperature in the set. Barriers that prevent complete sampling at low temperatures can be overcome readily at high temperatures.

After a sufficiently long REMD simulation, the trajectory for this replica will contain a correctly-weighted distribution of the conformations available at this temperature. This trajectory must be analyzed to group the structures sampled into distinct conformations.

Cluster analysis

The product of an REMD simulation is a trajectory for each temperature. For a sufficiently long simulation where the simulations were able to cross barriers freely, the configurations will be sampled according to their equilibrium probability. A discrete set of conformations must be identified from this trajectory. Cluster analysis can be used to identify discrete conformations in this ensemble by identifying groups of conformers that have similar geometries according to a chosen metric. Clustering works by assigning a metric to each configuration, measuring the distance between pairs of these configurations, and then grouping similar configurations into conformations based on this distance metric. Cluster analysis allow common conformations to be identified from the configurations of a trajectory using little to no a priori knowledge.

Work undertaken

In this paper, we present the implementation of a work flow for conformation searches using REMD and cluster analysis (see Fig. 1). This method supports conformation searches for molecules in the gas phase, implicit solvents, and explicit solvents. The method is implemented by integrating open source software using Python scripting. Examples of the conformation search results for two drug molecules are presented.

Figure 1 The work-flow for the conformation search method presented in this paper.

A parent script executes OpenBabel, VMD, and NAMD to generate the set of lowest energy conformations.

Theory

Replica exchange molecular dynamics

In REMD, m non-interacting replicas of the system are run, each at its own temperature, Tm. Periodically, replicas i and j exchange coordinates and velocities according to a criterion derived from the Boltzmann distribution (Earl & Deem, 2005; Mitsutake & Okamoto, 2000). In the implementation used here, exchanges are only attempted between replicas with neighboring temperatures in the series. Exchange attempts for replica i alternate between attempts to exchange with the i − 1 replica and the i + 1 replica. The exchanges are accepted or rejected based on an algorithm that ensures detailed balance, similar to the Metropolis criterion (Frenkel & Smit, 2002). By this criterion, the probability of accepting an exchange is, Pacc= min1,exp1kB1Ti−1TjV(ri)−V(rj)(3)

where V is the potential energy, and ri specifies the positions of the N particles in system i. A conformation is accepted/rejected if this probability is less than a random number between 0 and 1, which is taken from a uniform distribution. In a successful exchange, the coordinates of the particles of the two replicas are swapped. When the momenta of the particles are swapped, they are also scaled by a factor of TiTi+1 to generate a correct Maxwell distribution of velocities. The process of REMD is illustrated in the following pseudocode.

Cluster analysis

Configurations in the REMD trajectory are grouped into clusters that correspond to distinct conformations. The lowest energy conformation will correspond to the cluster with the greatest number of configurations. The process of clustering conformers involves using some proximity function to measure the similarity between pairs of conformations. This clustering algorithm groups these configurations according to the pattern proximity of this function (Jain, Murty & Flynn, 1999).

In this work, the solute root mean square deviation (RMSD) metric is used to identify the highly-probability conformations from the REMD trajectory. The RMSD provides a metric for the quality threshold of the similarity of two solute configurations. It is calculated from the Cartesian coordinates of the two configurations rk(i) and rk(j) each having N atoms using Becker et al. (2001), dij=[1NΣk=1N|rk(i)−rk(j)|2]1/2(4)

The quality threshold clustering algorithm groups objects such that the diameter of a cluster does not exceed a set threshold diameter. The number of clusters (N) and the maximum diameter must be specified by the user prior to the clustering analysis. A candidate cluster is formed by selecting a frame from the trajectory (a conformer) as the centroid. The algorithm iterates through the rest of the configurations in the trajectory, and the conformer with the smallest RMSD with respect to the centroid is added to the cluster. Configurations are added to this cluster until there is no remaining configuration with an RMSD less than the threshold. The clustered configurations are removed from consideration for further clusters and a new cluster is initiated. This process is repeated until N clusters have been generated.

Computational Work Flow

The first section describes a work flow that was developed to perform an explicitly-solvated conformational search of small drug molecules. In the second section, applications of the work flow are described, and the results are compared to gas phase and GBIS implementations.

Our method automatically performs conformational searches in the gas phase, implicit aqueous solvent, and explicit aqueous solvent for each solute structure. The work flow makes use of several open source programs, as illustrated in Fig. 2. The conformation search work flow can be divided into 5 steps.

Figure 2 Schematic of exchange attempts between four replicas simulated at temperatures T1, T2, T3, and T4.

After a large number of exchanges, each replica will have been simulated at the full range of temperatures. The lowest temperature replica will have contributions from each simulation.

Generation of initial 3D molecular structure.

Solvation of solute (for explicit solvent method only).

Equilibration MD simulation.

REMD simulation.

Cluster analysis.

Structure generation

The initial 3D structure is generated using the OBBuilder class of OpenBabel version 2.3.2. OpenBabel is a chemistry file translation program that is capable of converting between various file formats, but can also automatically generate 2D and 3D chemical structures and perform simple conformation searches (O’Boyle et al., 2011). Our work-flow uses OpenBabel to converts the SMILES string input, which is an ASCII string representation of a molecular structure, into an initial 3D structure that is saved in Protein Data Bank (pdb) format. OpenBabel supports many other chemical file formats, so alternative input formats can also be used. To generate a reasonable initial conformation, a conformer search is performed using the OBConformerSearch class of OpenBabel. This algorithm uses rotor keys, which are arrays of values specifying the possible rotations around all rotatable bonds (O’Boyle et al., 2012). Structures for each combination of rotor keys are generated and the potential energies for these conformations are calculated. The lowest energy structure for a rotor key is identified (Vandermeersch, 2006). Once all possible conformers have been generated, the algorithm selects the one with the lowest energy. The Generalized Amber Force Field (GAFF) is used for all OpenBabel MM calculations (Wang et al., 2004). Solvation effects are not included in this model.

One drawback of OpenBabel is that the current version can generate wrong stereoisomers for chiral centers in fused rings for some molecules. In these cases, the user should check the initial structure to ensure that the correct stereoisomers is modeled.

Solvation of solute

The Antechamber utility of the Ambertools suite is used to generate the necessary topology (.rtf) and parameter (.prm) files of the solute (Wang et al., 2006). This utility automatically detects the connectivity, atom types, and bond multiplicity of organic molecules and generates the parameter file and topology files based on the GAFF. The psfgen plugin VMD is used to generate a Protein Structure File (PSF) for the molecule from the RTF file. For simulations with an explicit solvent, the Solvate plugin of VMD is used to add a 10 Å layer of water in each direction from the furthest atom from the origin in that direction. This creates a periodic unit cell that is sufficiently large so that solute-solute interactions and finite-size effects are small. For ionic molecules, the autoionize VMD plugin is used to add Na+ or Cl− ions such that the net charge of the simulation cell is zero.

Equilibration

For simulations with an explicit solvent, MD simulations are performed with NAMD to equilibrate the system prior to the conformational search. For the gas phase and GBIS models, a 1 ns MD simulation using a Langevin thermostat is performed. For the explicit solvent simulations, a 1 ns isothermal-isochoric (NVT) simulation is followed by a 1 ns isothermal-isobaric ensemble (NpT) simulation A Langevin thermostat and a Langevin piston barostat are used to regulate the temperature and pressure of the system, respectively.

To simplify visualization and analysis, the center of mass of the solute is restrained to remain at the center of the simulation cell using a weak harmonic restraining force. This restraint is imposed with the Colvar (Collective Variables) module of NAMD using a force constant of 5.0 kcal Å−2.

Replica exchange MD

Using the equilibrated system, a replica exchange MD simulation is performed to sample the configurational space of the system. A total of 24 replicas are simulated, with a range of temperatures between 298 and 500 K. The temperatures of the replicas are spaced according to a geometric series (Kofke, 2004; Earl & Deem, 2005). A 1 ns equilibration followed by a 10 ns sampling simulation is performed for each replica. Configurations are saved and exchanges are attempted every 1,000 time steps. The REMD simulations were were performed at constant volume, which was the final volume of the NpT equilibration simulation.

Cluster analysis

The trajectory of the lowest temperature replica is analyzed by clustering analysis to identify the most probable conformations. The positions of the solute atoms in each frame of the trajectory are rotated and translated to minimize the RMSD. The cluster routine of the measure module of VMD is used to identify highly-weighted conformations. This routine uses the quality threshold clustering algorithm, with the RMSD as the metric. An RMSD cutoff of 1.0 Å was used. In this work flow, five clusters are generated. The clusters are sorted in order of the largest to smallest numbers of configurations included, the first of which is the most important as it represents the most probable conformer for the lowest temperature replica. The configurations that are part of each cluster are saved to separate trajectory files. The conformation is defined by the set of configurations grouped into this trajectory file.

Implementation and Usage

The work flow is implemented in a Python script that calls external programs and processes the data from these programs. This script is responsible for handling user input and integrating the work flow into the a PBS-type queuing system. PBS is a distributed workload management system, which is responsible for queuing, scheduling, and monitoring the computational workload on a system (Urban, 2010). The program is executed by the command,

python fluxionalize.py −p [number of processors, default is 2] −n [name, default is "test"] −l [location/directory, default is current working directory] −c [number of clusters to save in [name]_out per instance, default is 1] −i [input]

When the calculation has completed, the following files/directories will have been generated in the specified/default location: 10.7717/peerj.2088/utable-1 [name]_out	contains the conformer pdb files for each instance	
[name].out	the logfile from the queue containing all the runtime command line outputs	
[name].tar.gz	contains all the files used and generated by the work flow, compressed for space	

OpenBabel is used to parse the molecular structure provided by the user and convert it to an initial 3D conformation, so any of the input formats supported by OpenBabel can be used. The examples presented here use SMILES (Simplified Molecular Input Line Entry System) strings as the input. SMILES denotes chemical structure as ASCII-type strings. If using a SMILES string, the input for the fluxionalize.py script is in the form of −i ‘[SMILES string].’ For other files types, the input is in the form of: −i [file]. In this case, if no name is specified with the −n option, then the file name is used in its place.

Availability

The code and required source files are available freely from GitHub at https://github.com/RowleyGroup/fluxionalize.

Technical Details

The current version of this code uses OpenBabel 2.3.2 (O’Boyle et al., 2011) and VMD 1.9.1 (Humphrey, Dalke & Schulten, 1996). All MD and REMD simulations were performed using NAMD 2.10 (Phillips et al., 2005). Bonds containing hydrogen were constrained using the SHAKE algorithm (Ryckaert, Ciccotti & Berendsen, 1977). Lennard-Jones interactions were truncated using a smoothed cutoff potential between 9 and 10 Å. A Langevin thermostat with a damping coefficient of 1 ps−1 was used. The simulation time step was 1 fs. Generalized born model simulations used a dielectric constant of 78.5 and an ion concentration of 0.2 M. For the simulations with an explicit solvent, water molecules were described using the TIP3P model (Jorgensen, 1981). The molecule and solvent were simulated under orthorhombic periodic boundary conditions. The electrostatic interactions were calculated using the Particle Mesh Ewald (PME) method with a 1 Å grid spacing (Phillips et al., 2005). Isothermal–isobaric MD simulations used a Nosé–Hoover Langevin piston barostat with a pressure of 101.325 kPa, a decay period of 100 fs, and a oscillation period of 2,000 fs. The potential energy terms for the solute were described using GAFF. The total potential energy function for this force field is Wang et al. (2004), V(r)=Σbondskb(r−req)2+Σangleskθ(θ−θeq)2+Σdihedralsϑn2[1+cos(nϕ−γ)]+           ΣiΣi<j4ϵij[(σijrij)12−(σijrij)6]+qiqj4πϵo1rij(5)

where req is the equilibrium bond length, θeq is the equilibrium angle, kb, kθ, and Vn are the force constants, n is the multiplicity, and γ is the phase angle for torsional angle parameters. The last summation represents the non-bonded interactions, including London dispersion forces, Pauli repulsion, and electrostatic interactions. εij and σij are the Lennard-Jones well depths and radii for a given pair of atoms, and qi is the partial charge of atom i. Atomic charges are assigned using the restrained electrostatic potential fit (RESP) charge fitting method (Wang, Cieplak & Kollman, 2000), where the atomic charges were fit to the AM1-BCC model (Jakalian et al., 2000).

Examples

To demonstrate the capabilities and performance of our method, conformation searches were performed on two drug molecules: α-amanitin and the neutral state of cabergoline (Fig. 3) (Bushnell, Cramer & Kornberg, 2002; Sharif et al., 2009). α-Amanitin serves as a good example of the effectiveness of the work-flow. There are significant differences between the primary conformers in the gas phase, implicit solvent, and explicit solvent models. The most probable conformations derived from these models are overlaid in Fig. 4. The gas phase structure is more compact than the explicit solvent structure, which is consistent with the tendency of gas phase molecules to form intramolecular interactions, while solution structures can extend to interact with the solvent. The implicit solvent model structure is more similar to the explicit solvent structure, but is still distinct from the explicit solvent structure. Figure 5 shows the four most probable conformations from the explicit solvent simulations. The clustering algorithm successfully categorized conformations with different configurations of the fused rings and orientations of the pendant chains.

Figure 3 Chemical structures of molecules used to demonstrate conformation search work-flow.

(A) cabergoline and (B) α-Amanitin are mid-sized pharmaceuticals with significant conformational flexibility. The intramolecular and solute-solvent interactions result in complex conformer distributions.

Figure 4 Most probable explicitly solvated α-amantin conformers.

(A) is the most probable, (B) is the second most probable, and so forth.

Figure 5 Most probable α-amanitin conformers.

The explicitly solvated (A) and GBIS (C) conformers show the effect of the solvent, as compared to the more compact conformer in the gas phase (B).

Cabergoline has a simpler chemical structure, containing no long chains and a more rigid ring structure. The most probable conformers with the explicit solvent (see Fig. 6B) are all quite similar; the RMSD values are under 0.98. Significant differences are apparent in the primary conformers of the explicit, GBIS, and gas phase simulations (see Fig. 6A). In particular, the configuration of the alkyl chains are sensitive to the effect of solvation. Generally, more rigid molecules will likely be less sensitive to solvation effects.

Figure 6 The lowest energy conformations of cabergoline calculated using the implicit and explicit solvent models.

(A) Most probable conformers, where is the explicit solvent is blue, gas phase is red, and GBIS is grey. (B) Most probable conformations calculated using explicit solvent models. In order of most to least probable: blue, red, grey, orange.

Cabergoline contains two nitrogen centers that are formally chiral. Some conformation search algorithms have difficulty with type of moiety because the chirality of these centers can be switched by inversion of the nitrogen center. These inversion moves must be explicitly implemented into the structure generation algorithm of the method. Because the method presented here uses REMD, these inversions occur thermally, so conformations corresponding to these inverted configurations are identified automatically.

The computational cost of these simulations is moderate. The most computationally-intensive step is the REMD simulations in the explicit solvent. These simulations completed after approximately 80 h when run on 72 2100 MHz AMD Opteron 6172 processors. Although the computational resources needed for REMD conformational searches are considerably greater than for the high-throughput heuristic methods that are currently used in high-throughput screening, these calculations are currently tractable. As the cost of these simulations scales well, this type of simulation could become routine when computational resources are widely available.

The average acceptance rates for the exchanges in the REMD simulations are collected in Table 1. The acceptance probabilities of the gas phase and implicit solvent models were high (> 80%). REMD in an explicit solvent was found to be an efficient means to sample the configuration space, with acceptance probabilities of 27 and 31% for the simulations of α-amanitin and cabergoline, respectively. REMD can be inefficient for simulations in explicit solvents because the acceptance probability decreases with the heat capacity of the system, which is proportional to the number of atoms in the system (Lingenheil et al., 2009). For large molecules that must be enclosed in a large solvent box, a prohibitively high number of replicas would be needed to ensure a sufficiently exchange probability. For small and medium sized molecules, like the ones used here, the simulation cell is small enough so that the exchange acceptance probability is > 0.25.

Table 1 Acceptance rates of exchanges for replica exchange simulations, averaged over all replicas.

The gas phase and GBIS simulations have very high acceptance rates, but the explicit solvent simulations have much lower acceptance.

Molecule	Simulation	Average acceptance rate	
α-amanitin	Explicit	0.27	
Gas phase	0.83	
GBIS	0.84	
Cabergoline	Explicit	0.31	
Gas phase	0.88	
GBIS	0.88	

The initial coordinate (.pdb) files for the explicitly solvated structures, and for the gas phase and implicitly solvated structures can be found on the Github. Also available are the coordinate (.pdb) files for the four most probable explicitly solvated conformers (see Figs. 4 and 6B), the coordinate files for the most probable conformers in gas phase and implicit solvent (see Figs. 5 and 6A), and the SMILES strings for α-amanitin and cabergoline.

Conclusions

In this paper, we described a workflow for performing conformational searches using REMD and clustering analysis for molecules in the gas phase, implicit solvents, and explicit solvents. The workflow consists of five primary steps: generation of a 3D structure, solvation of the solute (for the explicit solvent method), an equilibration MD simulation, a REMD simulation, and cluster analysis. This method is implemented in Python scripting by integrating several open source packages (i.e., OpenBabel, VMD, and NAMD). The workflow makes use of the greater conformation sampling achieved by REMD, and then performs cluster analysis to find the most probable conformers sampled in the trajectory. Two drug molecules were used as examples of the work-flow, which show significant differences between conformers in the gas phase, implicit solvent, and explicit solvent. This work-flow has the potential to be applicable to many fields such as drug design, cheminformatics, and molecular structure studies.

Additional Information and Declarations

Competing Interests

Author Contributions

Data Deposition

The authors declare that they have no competing interests.

Kari Gaalswyk conceived and designed the experiments, performed the experiments, analyzed the data, contributed reagents/materials/analysis tools, wrote the paper, prepared figures and/or tables, reviewed drafts of the paper.

Christopher N. Rowley conceived and designed the experiments, performed the experiments, analyzed the data, contributed reagents/materials/analysis tools, wrote the paper, prepared figures and/or tables, reviewed drafts of the paper.

The following information was supplied regarding data availability:

GitHub: https://github.com/RowleyGroup/fluxionalize.

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
