# Peer review of "An explicit-solvent conformation search method using open software"

_PeerJ, doi:10.7717/peerj.2088_

## Round 0.1 · original submission · Minor Revisions

Besides addressing all issues raised by the reviewers, I would like you to consider redrawing Figure 4, which is very hard to parse due to the low superposition of the conformers. Wouldn't it be better to show each conformer separately?

·

Basic reporting

see below

Experimental design

see below

Validity of the findings

see below

Additional comments

In my opinion the following issues should be addressed before the paper is suitable for publication

1. The following Information needs to be provided (as text files on, e.g. Figshare ) for reproducibility:
a. The smiles strings used for Amanitin and Cabergoline.
b. Start coordinates used for simulations
c. Coordinates for the structures shown in Figures 4 and 5
d. In addition, the authors should describe the protonation state they used for Cabergoline

2. The authors may want to note in the manuscript (warn the reader) that Babel smiles conversion can produce the wrong stereo isomers for chiral centers in fused rings.

3. a. The authors may want to note that their scheme identifies the lowest free energy conformer rather than the lowest potential energy conformer (which is what oBabel and related methods do). B. related to this, a discussion of whether the simulation is run sufficiently long to yield meaningful free energy differences would be useful

4. In the Solvation of Solute subsection the authors mention the generation of CHARMM-format parameters, but later write “The potential energy terms for the solute were described using the General Amber Force Field (GAFF).” This was confusing to me and the authors should clarify.

5. Cabergoline contains two tertiary amines which represent particular challenges for conformational searches since they, in effect, are chiral centers and inversion is not considered a degree of freedom in the oBabel conformer search when these N atoms are in a ring. The starting “chirality” of these centers in the starting structure is thus more or less arbitrary. Furthermore, if the N is protonated this degree is certainly not sampled in the MD as it would effectively require deprotonation. For deprotonated centers it is also not obvious that the inversion barrier is low enough to allow this degree of freedom to be sampled. The authors should discuss this point.

Reviewer 2 ·

Basic reporting

"No Comments"

Experimental design

"No Comments"

Validity of the findings

"No Comments"

Additional comments

The manuscript entitled "An explicit-solvent conformation search method using open software" explain a new and novel workflow for conformational search. Before the publication some minor comments should be addressed:
1 - As the authors mentioned in the abstract "We have developed a
work-flow for performing a conformation search on explicitly-
solvated molecules using open source software.", they developed
a new workflow and not a new method. I recommend to change
the title to be more accurate to something like: An explicit-solvent
conformation search work-flow using open software
2 - In the introduction and other parts of the manuscript would be
important to cite some previous softwares and algorithms in
conformational search. For example, this recently manuscript
should be cited PMID: 23030613.
3 - Presumably , one of the major objetives of conformational search
is Drug design. It would be interesting to see in the manuscript a
table with a time measures for different molecules complexity and
set with different number of molecules.
4 - As I said, the manuscript do not do any reference to other
softwares that do conformational search. Would be interesting to
have some references and also in the Suppl information or
somewhere a short comparison with previous softwares like
OpenEyes, TFD.
5 - Is the current algorithm applicable to conformational search in
fields like drug design and docking.

---

## Round 0.2 · accepted · Accept

I deem the reviewer's comments to have been suitably addressed.

In the submitted revision the structures of cabergoline and alpha-amanitin in Figure 3 are swapped. In the legend of figure 4 "solvated α-amanitin conformers from most," should be "solvated α-amanitin conformers" . Please address these issues with PeerJ's technical staff during the production stage